# What outcomes are important to families with a lived experience of stillbirth? A qualitative study to inform the development of a core outcome set for stillbirth care

Danya Bakhbakhi[1]*, Christy Burden[2], Anna Davies[3], Abigail Fraser[4], Dimitrios Siassakos[5], Mary Lynch[6], Laura Timlin[7], James M. N. Duffy[8], Maggie Redshaw[9], Heatherjane Dangerfield[10], Alexander E. P. Heazell[11], Lisa Hinton[12], iCHOOSE Collaborative Group[¶]

**1** Translational Health Sciences, Bristol Medical School, University of Bristol, Bristol, United Kingdom, **2** Translational Health Sciences, Bristol Medical School, University of Bristol, Bristol, United Kingdom, **3** Translational Health Sciences, Bristol Medical School, University of Bristol, Bristol, United Kingdom, **4** Population Health Sciences, Bristol Medical School, University of Bristol, Bristol, United Kingdom, **5** Institute for Women's Health, University College London, London, United Kingdom, **6** Bristol Medical School, University of Bristol, Bristol, United Kingdom, **7** North Bristol NHS Trust, Bristol, United Kingdom, **8** Department of Women & Children's Health, School of Life Course & Population Sciences, Kings College, London, United Kingdom, **9** Brazelton Centre, Cambridge, United Kingdom, **10** Sands and iCHOOSE parent and public representative, London, United Kingdom, **11** Maternal and Fetal Health Research Centre, Division of Developmental Biology and Medicine, University of Manchester, Manchester, United Kingdom, **12** Nuffield Department of Primary Care Health Sciences, University of Oxford, Oxford, United Kingdom

¶ The complete membership of the iCHOOSE Collaborative author group can be found in the Acknowledgments
* Danya.bakhbakhi@bristol.ac.uk

## Abstract

### Objective

To identify outcomes that are important to families, to inform the development of a core outcome set for stillbirth care research.

### Design

Qualitative interview study.

### Setting

A national study in the United Kingdom.

### Population

A diverse sample of parents with a personal history of stillbirth were interviewed.

### Methods

Data collection, coding and analysis were influenced by a modified Grounded Theory approach. Parents' lived experiences of stillbirth were translated into outcomes for the purpose of developing a core outcome set.

**Data availability statement:** All relevant data are within the paper and its Supporting Information files. Full data cannot be shared publicly due to data contain potentially identifying or sensitive patient information and to maintain confidentiality. Data are available from the University of Oxford and Berkshire Ethics Committee (hergadmin@phc.ox.ac.uk) REC Ref 12/SC/0495. for researchers who meet the criteria for access to confidential data.

**Funding:** This report is an independent research arising from a doctoral fellowship (DRF) supported by the National Institute for Health Research (NIHR DRF-2017-10-130). The views expressed in this publication are those of the authors and not necessarily those of the National Health Service, the National Institute for Health Research or the Department of Health. There was no additional external funding received for this study.

**Competing interests:** DB is Trainee Scientific Editor for BJOG. AEPH is the Director of the Tommy's Stillbirth Research Centre, University of Manchester which receives grant funding for stillbirth research.

## Results

Forty parents and family members were interviewed. Analysis identified 349 potential care outcomes, 303 (87%) of which have not been previously reported by stillbirth care studies. Outcomes were organised into four major care outcome themes: 1) Clinical 2) Mental health and wellbeing 3) Social and family 4) Future pregnancy and children. Short- and long-term outcomes related to the labour, birth, investigations to understand why a baby had died, stillbirth in a multiple pregnancy, postpartum, psychological and subsequent pregnancy care were reported. Outcomes infrequently measured in previous stillbirth care research yet discussed by most participants were social isolation, impact on occupation and need for mental health support. Parents spoke of the importance of counselling to help them understand their grief, however, the provision of this service was reported to be varied throughout the UK.

## Conclusion

A comprehensive outcome inventory has now been constructed, from which the final core outcome set will be determined. Future care should be developed and evaluated using outcomes that directly relate to the lived experiences of parents and families exposed to stillbirth.

## Introduction

A core outcome set that is relevant to bereaved families is needed to advance the evidence for care after stillbirth [1]. A core outcome set is a consensus-derived minimum set of outcomes that should be measured and reported in all clinical trials of a specific disease or trial population [2].

A prior systematic review on interventions, outcomes and outcome measurement tools in stillbirth care research (defined as care provided after a stillbirth has occurred) identified wide variation in outcomes reported and considerable research evidence gaps for different interventions, such as bereavement counselling and specialised care prior to or during a subsequent pregnancy after stillbirth[3]. Outcomes reported in previous studies of stillbirth care, have primarily focused on medical outcomes, negating the wider impacts of stillbirth on parents [3]. For example, findings from the same systematic review found no studies on labour and birth care reporting psychological outcomes such as grief or anxiety [3]. Similarly, no studies evaluating bereavement hospital care considered outcomes relating to parents' role, for example, returning to work or caring for older children [3]. Consequently, it has not been possible to synthesise results and draw conclusions on optimal medical, psychological or supportive care for parents following a stillbirth.

Only ten studies in the systematic review included patient and public perspectives in their study design, suggesting that the outcomes identified are likely to reflect those that researchers believe are important to measure [3]. Parents who experience stillbirth may have different views from those of researchers and professionals

about which outcomes are important, leading to important outcomes being overlooked when evaluating interventions after stillbirth.

Previous qualitative studies including people with lived experience of a medical or pregnancy condition has been useful in identifying novel outcomes that might be relevant and important for a core outcome set [4–7]. There is a wealth of qualitative research on the psychosocial impact of stillbirth on parents and their experience of care afterwards [8,9]. However, no previous qualitative research has specifically looked at which outcomes should be measured in stillbirth care research, incorporating the views of mothers, fathers, birthing people, and non-birthing partners.

To address these limitations, a minimum set of outcomes that should be reported in all stillbirth care research studies is being developed with bereaved parents, researchers, healthcare professionals and charity representatives (iCHOOSE Study) [1]. This core outcome set will aim to address heterogeneity in outcome reporting, reduce outcome reporting bias, and allow interventions to be compared and results to be combined, thus strengthening the evidence available. The aim of this research was to capture the lived experience of stillbirth, to determine what outcomes could be important to measure for the development of a core outcome set for stillbirth care research.

## Methods

The core outcome set study was prospectively registered with the Core Outcome Measures in Effectiveness Trials (COMET) Initiative registration number 775 [10]. A protocol with pre-defined methods has been published [1]. This qualitative analysis forms part of the first stage in the development of a core outcome set for stillbirth care research, to help identify potential outcomes for a long-list to subsequently prioritise in a modified Delphi consensus survey (See Fig 1). There is no agreed set methodology for identifying outcomes using qualitative data from lived experiences, therefore data analysis was informed by previously published qualitative core outcome set development research [4,11].

### Ethics approval

Ethical approval was obtained from the Berkshire Ethics Committee REC Ref 12/SC/0495.

### Patient and public involvement (PPI)

Parents with lived experience of stillbirth and charity representatives were involved throughout this study to inform research design, recruitment, topic guide, analysis and write-up.

### Sample and recruitment of participants

Semi-structured narrative interviews were conducted with parents whose baby died before or during birth at 24 completed weeks' gestation or more, the legal definition for stillbirth in the UK. Mothers, fathers, birthing people, non-birthing partners and family members (bereaved siblings and grandparents) with a lived experience of stillbirth were invited to participate. A targeted advertisement was shared through stillbirth support charities, a hospital site in the South-West of England, PPI collaborators, and snowballing. Due to the method of recruitment, it was not possible to ascertain the total number of family members invited to participate and therefore a response rate or reasons for declining is not documented.

Interested family members emailed the research team to arrange a potential interview. DB (project lead) facilitated recruitment into the study. To ensure diverse opinions, participants were purposively sampled to gain maximum variation in their views and experiences. Parents who had experienced stillbirth at least six months prior to the study were eligible to participate. The research team felt that participation within six months of their baby's death might be too distressing for parents and responses potentially influenced by acute grief.

Interviews continued until it was considered that theoretical saturation was approaching (i.e., no new themes or outcomes emerged). [12]. To ascertain if the sample size was sufficient, dimensions of information power were considered

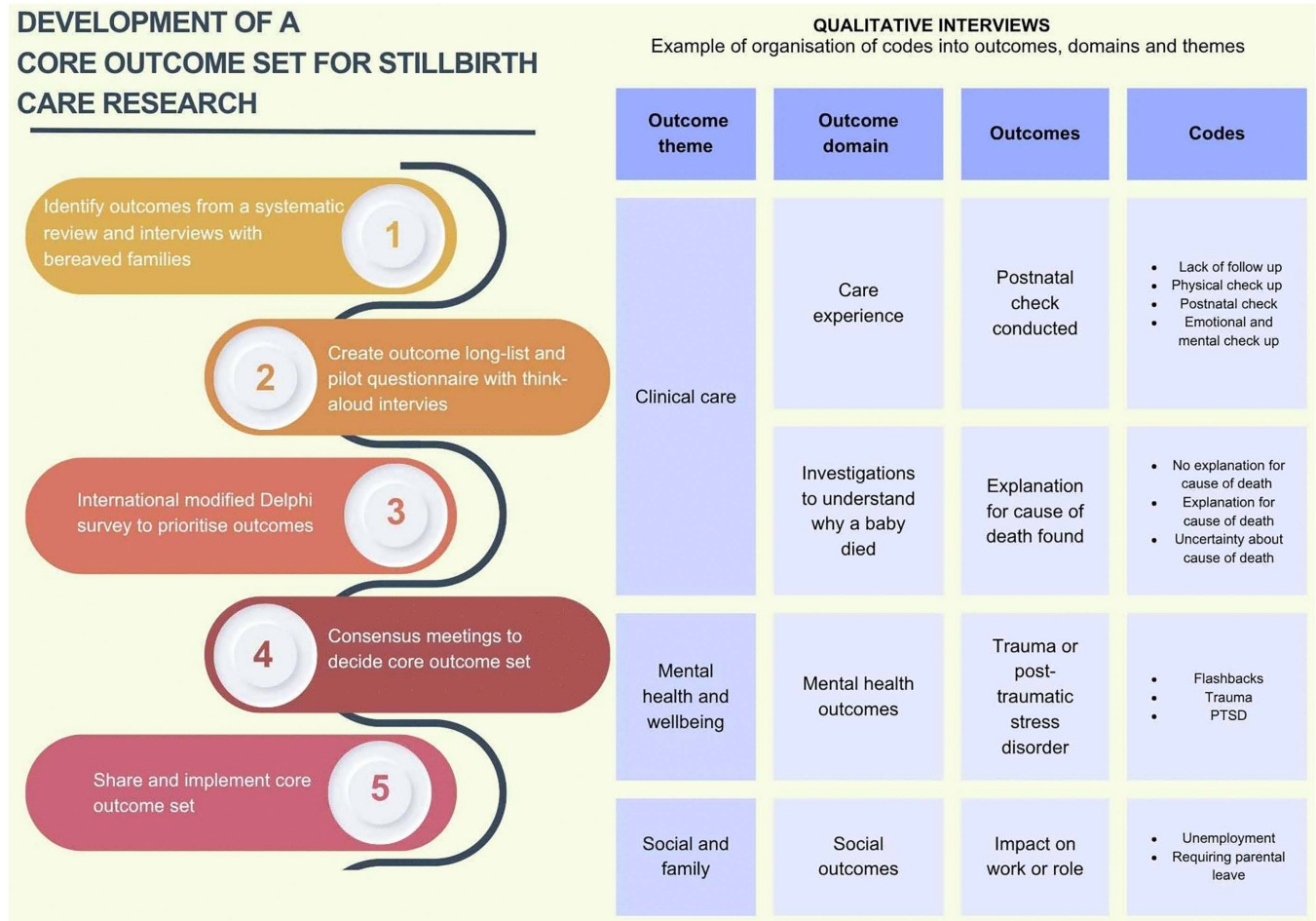

**Fig 1. Development process of core outcome set for stillbirth care research and organisation of codes into outcome.**

including study aim, sample specificity, use of established theory, quality of dialogue and analysis strategy [13] (S1 Table). Coding was managed using NVivo 12.0 software [14]. Following coding of the 25th interview, a review of the codebook and reflexive conversations within the research team took place. It was agreed that no new major categories of outcomes were emerging. However, to ensure maximum variation, three additional sets of parents who had experienced stillbirth in a multiple pregnancy were interviewed.

## Data collection and analysis

Parents were interviewed individually or jointly (with partner or family member), according to preference between 1st September 2018 and 30th June 2019. After obtaining informed written consent, face-to-face interviews with parents were conducted in either their own homes or a suitable private location of their choice. The lead researcher (DB) had training in qualitative interview methods and conducted the interviews supervised by an experienced qualitative researcher (LH). A second researcher was present at each interview (CB, ML, LT, LH) to support the lead researcher. Interviews were in two parts; parents were invited to first narrate their lived experience of stillbirth, followed up with semi-structured interview questions using a flexible topic guide to aid discussion of care outcomes (S2 File).

Interviews were audio and/or video recorded with parents' consent. If a parent became upset while narrating their lived experience of stillbirth, we offered the opportunity to pause or stop the interview. Parents were signposted to community support. The recording was transcribed by a professional transcription service. All transcripts were checked for accuracy and anonymised. Transcripts were sent back to all participants to check content accuracy.

During the initial interviews it became apparent that the word 'outcome' was somewhat abstract or nebulous to some participants [15]. Furthermore, it was often difficult to disentangle parent experiences, their impact, and frame these as outcomes that should be considered in future research assessing stillbirth care. Therefore, specific examples were used from parents' personal interview accounts of their care, to help identify potentially important outcomes (e.g., "You mentioned you had no hospital follow-up– how did that affect you?"), and the reported lived experiences and impacts of stillbirth, were used by the researchers to develop outcomes that could be of importance.

Data collection was influenced by a modified Grounded Theory approach [16], where analysis of initial interviews enriched data collection of later interviews. A familiarisation process was undertaken, whereby each transcript was read in conjunction with listening to every audio-recording. Each transcript was coded in duplicate with a second experienced qualitative researcher (LH or AD) coding blinded and in parallel with DB. Each line of the transcript was coded to identify potential outcomes from reported experiences using an inductive technique. Selective coding took place to understand larger codes and identify specific outcomes using the 'One sheet of paper' technique for coding [17]. Codes were grouped (where relevant) and experiences translated into outcomes through discussion with the research team. Similar outcomes were arranged into outcome domains, which were then organised into four overarching care outcome themes. Each participant's contributions/outcomes, whether from the bereaved mother, partner or family member were coded individually and each quotation presented in the results is attributed to the specific parent who said it.

See Fig 1 and S3 Fig. Outcomes generated were cross-referenced against the outcomes extracted from our systematic review on interventions and outcomes in stillbirth care research [3]. The number of novel outcomes identified from the interviews was recorded. These were then added to the long-list of outcomes for consideration in the next steps of the core outcome set development; a pilot with think-aloud interviews and a modified Delphi consensus survey (to be reported in a subsequent publication).

A quality framework proposed by Yardley et al (2000) was applied, to demonstrate trustworthiness and rigour (S4 Table) [18].

## Results

Twenty-nine interviews took place between September 2018 and June 2019. Thirty-eight parents (26 mothers and 12 fathers) and two bereaved siblings were interviewed. These interviews included 26 families, including 16 individuals and 12 couples (one family was interviewed on two occasions as the first interview was cut short due to an emergency). See Table 1 for detailed characteristics of participants.

Sixteen outcome domains were developed which were organised into four major care outcome themes; 1) Clinical care; 2) Grief and psychological; 3) Social and family 4) Future pregnancy and children. Three hundred and forty-three outcomes were identified from the analysis of the transcripts. Fig 2 illustrates the outcomes that are important to families after stillbirth.

The following results describe reports of the lived experience of stillbirth and impacts of care which were linked by the research team to agreed outcomes of importance. Table 2 exemplifies outcome domains and themes with illustrative quotes. See S5 Table for the outcome long-list derived from the interviews and S6 File for additional interview quotes.

### Theme 1: clinical outcomes

**Diagnosis, labour and birth outcomes.** Parents recalled the immense shock they felt when they found out that their baby had died. How the diagnosis was delivered to them by healthcare professionals was reported to have a significant

**Table 1. Characteristics of participants.**

| Circumstances of stillbirth | |
|---|---|
| Gestation of stillbirth, *median weeks (range)* | 37 (24-41) |
| Time of interview since stillbirth, *median years (range)* | 3.5 (1-52) |
| Stillbirth in a multiple pregnancy, *n* | 3 |
| Parity, *n* | |
| Nulliparous | 20 |
| Multiparous | 7 |
| Ethnicity, *n* | |
| White British | 31 |
| Minority ethnic group[a] | 9 |
| Subsequent pregnancy after stillbirth, *n* | 22 |
| Mothers, *n* | 26 |
| Fathers, *n* | 12 |
| Bereaved sibling, *n* | 2 |

[a]Asian Indian, Mixed White Asian or Black Caribbean and White European (other than White British)

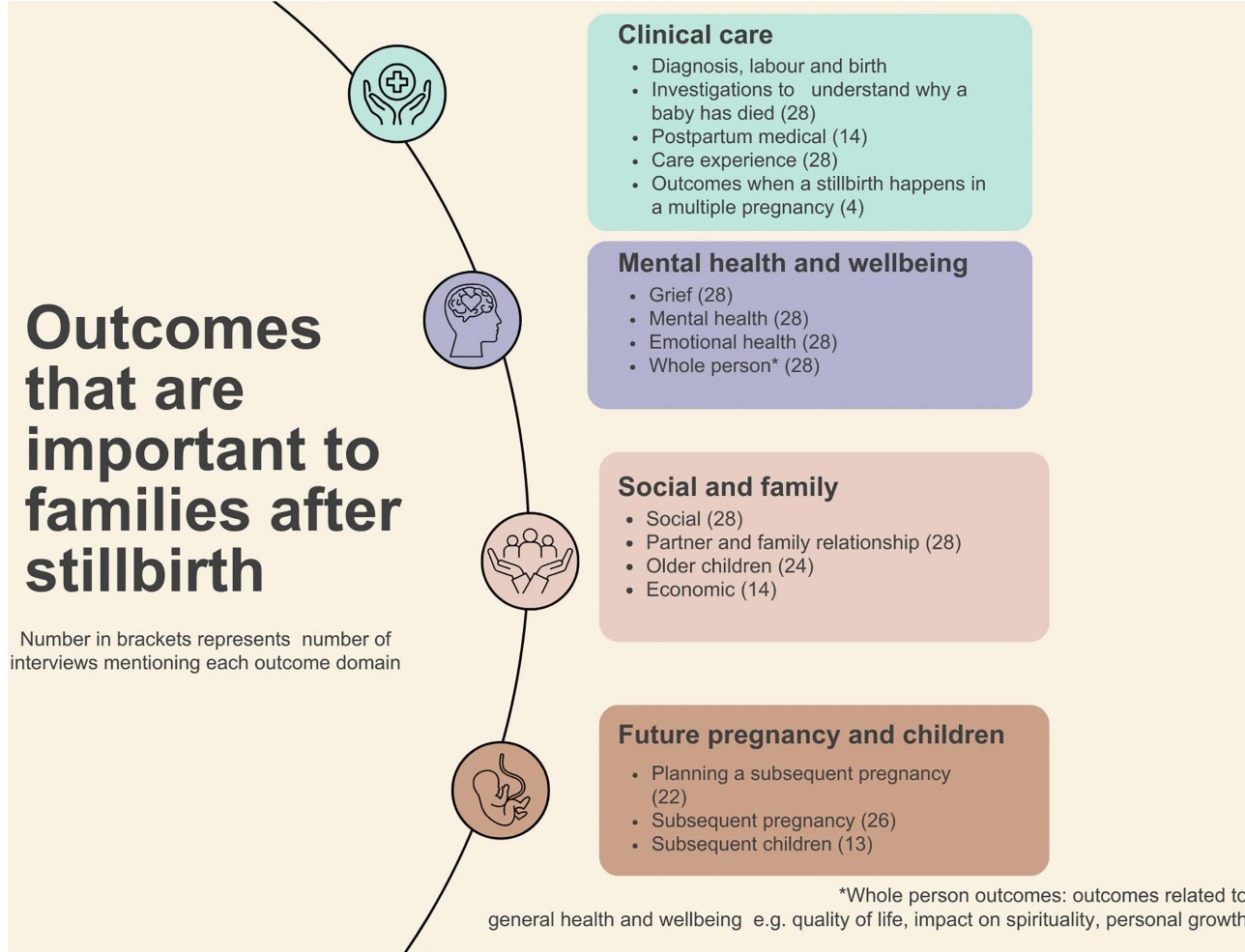

**Fig 2. Outcomes that are important to families after stillbirth.**

**Table 2. *Outcome themes and domains with illustrative participant quotes.***

| | | |
|---|---|---|
| Clinical care outcomes | Diagnosis labour and birth | **STB02 (Father):** "So the registrar was still scanning at the time… it was on like a retractable cord the scan, just let it go and it flew back into the machine, looked at us and went I'm sorry and left the room, just walked straight out the room." <br> **STB12 (Mother):** "we think actually she might prefer – it is like the final thing she can do for her baby and she might prefer to do it like this. And at that stage – so I did end up having him naturally." <br> **STB20 (Mother):** "I just lay for hours and then I got an epidural, because I just didn't want to feel anything. I was always really set on having a natural birth." <br> **STB18 (Mother)**: "Yeah I'm planning to have an elective caesarean this time. I think partly because I got pregnant so quickly; I just feel….it was really traumatic." |
| | Postpartum medical | **STB05 (Mother):** "It affects your brain. And even though you've not got your baby there, you still feel all the hormones, all the emotions that are waving all over the top of you after you've given birth, and there's no one there to tell you that any of this is normal. Even things like your boobs, milk, you just feel as if there's nobody there for you to give all this to, and there's no-one there to explain to you what it is you're going through, what these emotions and physical symptoms are." |
| | Outcomes related to investigations | **STB14 (Mother):** "I blame myself a lot and looked at things I could have done wrong that maybe caused it. I think that's why a post- mortem's important. I know some people choose not to have one but for me it was very important to know what went wrong." |
| | Outcomes when a stillbirth occurs in a multiple pregnancy | **STB28 (Mother):** "She came round and said would you like to get the boys together, would you like an opportunity to get the twins together. I would never have thought of that, why, he has died, we don't have twins, but she sort of said there is a space come free in the cot next to [Surviving twin] and if you want, I can put a screen up and we can bring him round and you can maybe just be together." <br> **STB26 (Mother):** "We were stressed at first. I think we spent three months constantly checking he was breathing so we were more stressed. That was the first three months and then we were also monitoring his development as well and he's only seven months and we're still looking if he is reaching the right milestones… Our main concerns are the baby's development. That's the most important for us." |
| Grief and psychological outcomes | Grief | **STB18 (Mother):** "I think having a fixed time in the week to express my feelings about things and what's going on in my life, and it's a sort of protected time when I know that I can do that and focus on grief." <br> **STB19 (Mother):** "But obviously once I'd had [Subsequent baby] and then you just get this surge of emotions again and we'd had a really rough ride carrying her, it was just like - I just couldn't keep the lid on top of the box any longer." <br> STB02 (Father): "I was a typical man in that I tried to keep everything together, do things practical, get back to work, get everything normal – what we thought was normal, back into the house. The issue with that was I didn't grieve properly myself and then subsequently about a year on then I had started having anxiety and panic attacks." |
| | Mental health | **STB07 (Father):** "So yeah it was through the GP (counselling) and I just started, I was sad and not clinically depressed or anything but you know I thought something's definitely got to change with my mental health." <br> **STB16 (Father):** I saw the GP just before my marriage broke up and, again, it was my ex-wife dragging me there, saying, "Can you fix him?" …he gave me some antidepressants. I didn't want to take them… I suddenly thought I needed a counsellor" |
| | Emotional | **STB20 (Mother):** "[If I had support] I just think I could have processed it better and I could have spent less time thinking I wasn't normal. It took me nine months to realise the feelings I was feeling were normal, when I was jealous of pregnant women and stuff, I actually thought I was the worst human being ever... I spent so much time thinking I was abnormal, and I was a horrible person." <br> **STB06 (Mother):** "I think having had a loss, like a stillbirth and then trying to conceive again, we felt very, very alone in that and it's been emotionally so draining not getting further along and I think– there's no one there that sits between this and the NHS." |
| | Whole person | **STB09 (Mother):** (When asked directly about care outcomes) "You almost feel like you've got so many different elements to this, to what's going on, whether it be how you feel about your family, how you feel about future pregnancy, how you feel about [Baby's name] herself, or the funeral or all these different things, or anniversary." |
| Social and family outcomes | Social outcomes | **STB23 (Mother):** "When you're knocking around at home suddenly and nobody's looking after you… you feel really let down and very lonely, very lonely." <br> **STB15 (Mother):** "No, there was just nothing, nothing on offer that gave me the opportunity to talk about it." <br> **STB10 (Mother):** "I was so anxious about going back that I eventually decided that I was worse off being at home than I would have been at work, …That was a massive step to begin with because I was going back to the hospital where [Baby's name] had died; even just walking through the doors of the hospital was really quite difficult…Then I did eventually see occupational health who organised for me to have a phased start over about a month." |

*(Continued)*

**Table 2.** (Continued)

| | Partner and family relationship | **STB14 (Mother):** "So it was like I could see how it breaks some people apart, going through this. It's brought us closer together, we've got a better understanding of how we're different, but we help each other in different ways to get through it." <br> **STB17 (Mother):** "He was much worse than me, I mean, I think he gave me a reason to get up and he said I did him. He was terrible, he was really bad, he was really aggressive and drinking a lot, which he wasn't before, and when he'd drink he'd get really angry. He was never angry to me, but I just didn't want that around me." |
| | Outcomes related to children | **STB27 (Father):** "I'm saying there was no guidance as to how you should [involve children]. I think there was an immediate attention for [Mother's name], very little on me and very little about how we then cope with our day to day lives, about kind of working forward with everybody. <br> **STB04 (Mother):** "So, my daughter had a baby four years ago and then she didn't want to have the baby at Hospital name], even though it was nearer, she went to XXX as she couldn't because the thought of you know… the legacy of that, the impact of that goes on for a long, long time, generations potentially." |
| | Economic | **STB22 (Mother):** "I think I could have done with more [counselling], but at the time we didn't have the money to pay for anymore and that's all I could get for free through the health insurance I had, so I just cut my losses and that was better than nothing." |
| Future pregnancy and children outcomes | Planning a subsequent pregnancy | **STB14** (Mother): "I'd like reassurances that my body can carry a child to term. Because for me, this is the only outcome I've ever known from a pregnancy… that's why I think I'd like help with fertility checks. I want reassurance really, to know that everything's okay. That would be helpful." <br> **STB01 (Mother)**: "No the first one (consultant) didn't make me feel confident we'd ever have another baby at all. She said that there's a 25% chance that this will happen again." |
| | Subsequent pregnancy | **STB08 (Mother):** "A live baby. That's all I wanted. I didn't care about anything else. My birth plan said I want my baby alive. That was it." <br> **STB10 (Mother):** "I really kind of wished that people had come in and said, I see that you've had a previous baby who died, I'm really sorry to hear that and I can see why you're anxious about this now." <br> **STB25 (Mother):** "The same person will do it each time…. Like I remember the first time having a scan there, I was so excited, which the only time in the pregnancy I felt anything close to joy… there was no rush, we were in there for an hour…and it's like nowhere on the NHS can you sit with someone for an hour." <br> **STB03 (Mother):** "So, I'm currently 24 weeks pregnant, so it's a lot of anxiety, but having a catch-up once a month is very useful for me, so, to go through the process of and managing the anxiety, it's very useful." <br> **STB11 (Mother):** "Awful, the absolute worse pregnancy on earth. I wanted her to move constantly, and she was a very quiet baby which was a nightmare, because I did have to go to hospital a few times in a panic saying, "She's not moving." They scanned me every two weeks and they were very good." |
| | Subsequent children | **STB09 (Mother):** "[Subsequent baby] is amazing, but it was really hard and it took to four or five months to bond with [Subsequent baby] and I just kept thinking he was somebody else's baby and he was going to die." |

long-lasting psychological impact on some parents. Mode of birth and induction of labour were often discussed. Nine mothers felt empowered to give birth to their baby vaginally and reflected on the experience positively. However, for others, the memory of the birth was distressing leading some to requesting a caesarean birth in a subsequent pregnancy. Having effective pain relief for their labour and birth was described as important, yet the side effects, such as drowsiness, nausea and vomiting needed to be considered.

**Post-partum medical outcomes.** Fourteen mothers described postpartum medical complications following the birth of their baby including postpartum haemorrhage, infection and anaemia needing treatment. The postnatal recovery including lactation management, in the absence of a baby, was vital, as mothers experienced the same hormonal and bodily changes as after a live birth. Parents felt having postnatal follow-up was a critical component of their care; however, the appointment was often not conducted by their care providers, thus having a negative impact on outcomes such as their mental health.

**Care experience.** Parents' experiences of clinical care after stillbirth varied; positive experiences included having individualised, compassionate, timely and supportive care. Having choices or options in their care was viewed positively by families. Parents appreciated when professionals respected the baby's existence, by offering opportunities to create memories (e.g., photographs and footprints). When this did not occur, parents described feeling like the baby did not

belong to them or was the "hospital's property". Midwives and doctors were viewed as having a crucial role in how parents felt about their identity and parenthood.

Negative experiences included inappropriate care settings and insensitive communication from healthcare professionals. In some cases, substandard care led to formal complaints and distrust in the hospital and medical profession; some sought second opinions or changed care providers in a subsequent pregnancy.

Five parents were concerned about the impact of providing stillbirth care on healthcare professionals; one parent described a midwife handing in her notice, and another recalled a student midwife "crying floods of tears" after her baby was born.

**Investigations to understand why a baby has died.**  Parents discussed the value of having or not having investigations to understand the cause of death (for example post-mortem, placental examination and blood tests). For all parents interviewed, finding out whether there was an explanation for the stillbirth, and what it was crucial. Respondents believed that the investigations could facilitate future pre-pregnancy counselling, inform management in subsequent pregnancies and prevent stillbirth recurrence. Results needed to be explained in a way that parents understood, and care providers not communicating these effectively and sensitively led to feelings of anger and guilt in the families.

**Outcomes when a stillbirth occurs in a multiple pregnancy.**  Three sets of parents who had experienced a stillbirth in a multiple pregnancy and a surviving twin sibling were interviewed. Outcomes that emerged included survival of the remaining twin and premature birth. Neonatal and childhood outcomes were identified, including the surviving twin reaching appropriate neuro-developmental milestones. Experiencing care that recognised the stillbirth had occurred and that the baby was still a twin was viewed as critical.

### Theme 2: mental health and wellbeing outcomes

**Grief outcomes.**  All parents experienced grief, and every parent's grief was individual. Whilst differences in the way men and women grieve were noted, common outcomes were identified. Grief outcomes included: feelings of overwhelming grief; unresolved grief; unrecognised grief and physical symptoms of grief. Common feelings related to overwhelming grief were shock, despair, emptiness, pain, feeling lost, hopelessness, anger and denial. As part of their grief many parents, described feeling self-blame, guilty and a sense of failure. Mothers described blaming themselves for the death of their baby or wondering if they could have prevented the stillbirth and so the investigations to understand why a baby died were instrumental in helping to resolve these feelings. Fathers described keeping their feelings to themselves or dealing with their grief at a later point compared with their partners, as they focused on practical tasks such as arranging the funeral and supporting their family. One father did not "grieve properly" in the first year which led to panic attacks and physical symptoms. Fathers often were not offered support by healthcare professions or did not seek support until many years later to help process their grief.

**Mental health outcomes.**  The mental health impact of stillbirth care was significant. Mothers and fathers described a diverse range of long-term consequences of stillbirth and poor care including depression, anxiety, post-traumatic stress disorder (PTSD), suicidal ideation, increased alcohol use and needing mental health support. Types of mental health intervention received included cognitive behavioural therapy, mindfulness, pharmacological treatment and eye movement desensitisation and reprocessing (EMDR) for PTSD. Parents described a 'postcode lottery' regarding access to perinatal bereavement counselling and some were surprised when this was not offered or part of standard care. Experiences of counselling were discussed by most parents and reported to have many mental health benefits, providing a space to talk about their experience and helping parents to understand their grief. Mental health support for fathers was variable and some described seeking support much later which had an impact on mental wellbeing. One father thought he was "paying for it now" as he let things build up and needed counselling therapy to talk and improve his emotional health.

**Emotional health outcomes.**  Stillbirth care influenced parents' emotional wellbeing and sense of control in their lives. Mothers described feeling embarrassed or shame whilst in hospital, when they had to be cared for alongside women

with living babies. Negative emotions such as stress, resentment, regret and low self-esteem were reported. Conversely positive emotions were also described; such the sense of pride they felt giving birth to their baby and seeing them for the first time. One parent recalled feeling more optimistic after psychological therapy, and another had more hope in a subsequent pregnancy when they had specialist care from their hospital.

**Whole person outcomes.** This domain describes a sub-set of outcomes that were recounted related to the general health and wellbeing of families, for example, quality of life, impact on spirituality or personal growth. As a direct result of their care or experience, many parents described participating in altruistic activities to help others who have experienced stillbirth such as peer support groups, raising money for charity, improving hospital care and taking part in research.

### Theme 3: social and family outcomes

**Social outcomes.** The social implications of stillbirth and care included profound perceived stigma and isolation. Parents spoke of feeling they had been abandoned, in an isolated part of the ward or bereavement suite, which was exacerbated by the lack of communication from healthcare professionals. Some felt they were a "lesser priority", due to staff being occupied with other women in labour. The feeling of isolation was amplified further by the lack of follow up and psychological support for parents when they left hospital. Several coping interventions were utilised by parents, to help alleviate the feeling of isolation, such as counselling and peer-support from other parents who had experienced stillbirth.

Support with returning to work and impact on job role were identified as important; parents discussed various care that supported their return to work including use of occupational health, work mentoring and counselling therapy.

**Partner and family relationship outcomes.** Perceived social support from those close to the parents was important, for example partner, family members and friends. Parents reported varying degrees to which their relationships with their partner were affected following the stillbirth. Mothers and fathers described the death of their baby strengthening their relationship and bringing them closer together as a couple. One mother felt that she and her husband dealt with things differently to begin with, however talking about their baby girl helped their relationship, this was facilitated by relationship counselling for three parents. Negative impacts on relationships were described in interviews such as more arguments, separation or divorce. Trying to become pregnant again after the stillbirth led to further challenges and "pressure" to some relationships, either when it took longer than expected to conceive or becoming pregnant too soon for one mother.

The impact on relationships in the family was important. According to those interviewed, family members such as grandparents and siblings grieved for the baby and found the emotional impact difficult. Having support and a family network helped many parents with their grief, however some parents described family members not wanting to talk when their baby died, which was challenging. Friends were also important as a source of support in challenging times and many appreciated people wanting to spend time with them and talk about their stillborn baby. Although friends and family wanted to offer support, sometimes they did not know what to say or do. This left some parents feeling very alone. Some parents described losing friendships when friends were not there for them. Others gained friendships after the stillbirth or found comfort in friends who had also experienced the loss of a baby in pregnancy.

**Outcomes related to older children.** The narratives described three impacts related to children; impact on older children; impact on parenting; and perceived support for children. Six parents had older children at the time of the stillbirth. All parents felt it was important that their children were aware of what happened. However, they differed in their preferences on whether to introduce their baby to their children. Four parents decided to not allow their children to meet the baby, either due to the overwhelming shock or the perception that the children were too young at the time to cope, whilst other older children, decided to meet the baby out of their own choice. Including the children in the memory making process, allowing them to hold, spend time and take photographs was a positive experience for one family, however their six-year-old was upset when they had to say goodbye to their sibling.

Two parents reported no significant effects of the bereavement on their children, whilst four described more long-lasting negative impacts on their mental health and behaviour, including separation anxiety and stress. In the short term, some

parents described finding it difficult to concentrate on their children. Whilst for others, older children provided a distraction, allowing some parents to keep going despite their grief. Parents wanted to try and keep everything as normal as possible for their children, by keeping their daily routines. Parents described themselves as being more anxious, cautious or over-protective following the stillbirth and described seeking support for children in a variety of ways. This included help from healthcare professionals such as the bereavement midwife, child bereavement charities, counselling services, school or support books written to help parents discuss the stillbirth with their children.

**Economic outcomes.** Impact on finances was recognised as an outcome that mattered. Although care provided by the hospital was free at the point of use, parents discussed the financial implications of arranging the funeral and psychological counselling. Parents spoke of often having to pay privately for their counselling or therapy either themselves or through their employer.

### Theme 4: future pregnancy and children outcomes

**Outcomes relevant to planning a subsequent pregnancy.** The decision about whether to attempt to become pregnant again was individual to each family. Advice about future management of subsequent pregnancies was central to many follow-up appointments after the stillbirth. Being told the risk of the stillbirth recurrence was identified as an outcome of importance. For those parents who wanted to become pregnant again, it was important to be supported by healthcare professionals and have some reassurances about future fertility and pregnancy risks. Four parents described having difficulty becoming pregnant again, requiring additional fertility treatment.

**Outcomes relevant to subsequent pregnancy.** Twenty-two parents went on to have subsequent pregnancies. Four parents had early miscarriages, however, there were no stillbirth recurrences. Once becoming pregnant, the primary concern was the health and survival of the baby. Parents discussed the timing and mode of birth in a subsequent pregnancy, as many did not want to go beyond their due date or the gestation at which their baby had died for fear of having another stillbirth. Eight parents were offered an induction of labour or caesarean, most often before full term (before 40 weeks) and two parents had preterm births (before 37 weeks' gestation) following induction either due to medical complications or parental preference following discussion with their consultant.

Parents valued continuity of care and where healthcare professionals acknowledged the previous stillbirth in a subsequent pregnancy. Outcomes related to hospital resource use in a subsequent pregnancy, such as requiring additional antenatal appointments, tests or ultrasound scans were identified. Two sets of parents were able to access a specialised clinic for subsequent pregnancies after stillbirths. Having individualised, supportive and specialist care was viewed as being helpful for mental health in a pregnancy after stillbirth. Anxiety was commonly described, and bereaved parents sought additional support from specialist antenatal classes, yoga, peer support groups and counselling.

**Outcomes relevant to subsequent children.** Outcomes related to subsequent children were discussed such as difficulties in attachment (bonding) to the baby, psychological impact on the subsequent child, and perceived support for parenting. Some mothers described a delayed grief reaction or postpartum depression which made it hard for them to bond with their new baby in a subsequent pregnancy. A few mothers recalled feeling very frightened and worried about whether their baby would survive the newborn period because of experiencing a stillbirth in the past. This led to more attendances at the doctors due to anxiety over their baby's health. Having empathic support with parenting from care professionals such as the health visitor was seen as crucial to help them cope.

**Comparison with outcomes reported in stillbirth care research.** We previously conducted a systematic review and mapped outcome reporting for stillbirth care research [3]. In total, the qualitative interviews reported here identified 349 distinct outcomes, 303 (87%) of which had not been assessed in the studies within the systematic review [3]. See S5 Table.

Outcomes rarely measured in previous stillbirth care research yet discussed by most participants were isolation, impact on occupation or role and need for mental health support [3]. The interviews identified additional outcomes to consider

measuring in relation to grief, and impact on relationships with the wider family including children. Seven new outcomes related to planning a subsequent pregnancy and fertility and 79 further outcomes related to care in a subsequent pregnancy were identified. Forty-one outcomes relevant to a stillbirth in a multiple pregnancy arose from the analysis.

## Discussion

### Main findings

Capturing the lived experience of parents using in-depth interviews revealed a much broader range of impacts compared with those previously reported in studies evaluating stillbirth care research. To fully facilitate evidence synthesis to inform what care after stillbirth is optimal, future research may need to focus on the additional outcomes identified in this study, such as impact on returning to work, isolation and mental health in a subsequent pregnancy.

### Strengths and limitations

Although there has been a wealth of previous research on the lived experience of stillbirth, this novel study has used in-depth analysis to identify impacts that can be used to distinguish outcomes relevant for studies of stillbirth care. A multi-disciplinary steering committee was established to guide the conduct of the study. Effort was made to ensure diversity for example varied parental experiences, geographical location and socio-economic backgrounds.

An important strength of this study was that an inclusive approach was adopted, including both parents and participation of other family members in the interviews, thus broadening the scope of outcomes identified. Similar outcomes were reported; however, fathers often spoke of the focus being on the mother in the early days and the lack of professional and social support after the death of their baby. This led to negative mental health outcomes and physical symptoms of grief for some fathers. Future research should focus on bereavement care interventions that include fathers as participants to improve psychological wellbeing. Some interviews were conducted together with the partner, this may or may not have influenced what a parent chose to share or emphasise, and we note this as potential limitation.

The interviews identified a wide range of impacts of stillbirth which is a testament to the rich data and rigorous analysis. Attempts were made to recruit parents from different ethnic backgrounds; however, not all minority ethnic groups were represented. For pragmatic reasons and due to resource limitations, this study was only conducted in English and in the UK. In many low- and middle-income countries, routine bereavement care after stillbirth is an emerging concept and parents' priorities of important outcomes may differ. To address this limitation international stakeholders have been given the opportunity to suggest additional outcomes later in the development process of the core outcome set for stillbirth care research [1].

Interviews were conducted approximately three and a half years after the primary event, with longest time since the stillbirth being 52 years. A possibly limitation is recall and the differences in care experienced by parents at that time, specifically the lack of bereavement care, counselling and support. Despite this many similar outcomes emerged from the analysis for example grief, stigma and isolation. By taking a longer-term perspective, the interviews were able to capture outcomes related to returning to work and future pregnancies [19–31].

Previous studies to develop core outcome sets on maternal health have utilised qualitative methods to inform the outcome long list, and they too have found additional outcomes not previously measured in research [4,32]. However, qualitative research is intensive and requires a substantial amount of time and investment. There could be more efficient ways of identifying outcomes, for example, systematic reviews and synthesis of the qualitative literature, focus groups or secondary analysis of qualitative data, although these methods may rely on data that is less focused on the researchers' primary objectives.

### Interpretation

The interviews have identified a broad range of impacts of stillbirth, many of which have not previously considered as study outcomes by stillbirth care researchers. All parents interviewed experienced grief in one form or another, yet their

lived experience of this grief was individual. Grief alone is not a unique outcome, and several studies have measured grief when evaluating care after stillbirth [3,33–38]. However, what was most interesting was the multitude of facets of grief that could be measured as identified from the analysis of the interviews, for example overwhelming grief, coping with grief, feelings self-blame, guilt and failure. Any number of these could be measured in a future study, and as such we need to understand what is most important to measure from the perspectives of parents, healthcare professionals and researchers, so as we are able to use the correct measurement tools at an appropriate time.

Social isolation and stigma were important impacts experienced and hence should be considered in future research. A Stillbirth Stigma Scale has recently been developed to measure this important outcome [39]. The finding of parents who have experienced stillbirth feeling isolated is not new [40–48]. For example, parents who suffered perinatal bereavement during the Covid-19 pandemic in the UK found that lockdown restrictions and changed maternity care left them feeling particularly isolated [49]. In the wider context, social isolation and loneliness can lead to a number of negative health consequences such as high blood pressure, mental health conditions and declining cognitive function [50–52]. It is therefore imperative that bereavement support exists to minimise these negative effects and that when assessing interventions post-stillbirth, this is measured.

Parents spoke of the importance of counselling to help them process their grief and provide them with a space to talk about their baby and their stillbirth experience. However, the provision of this service appeared to be inequitable throughout the UK. In 2021, the NHS began to implement more mental health provision for perinatal bereavement nationally and so availability of this care may have improved. Despite this, the evidence for the effectiveness of counselling as an intervention after stillbirth is limited, [33,35,53,54] as previous studies have measured different outcomes, so we are unable to compare and combine studies in meta-analyses. A core outcome set for stillbirth care could address this issue in future clinical trials.

Outcomes related to the whole family are essential. For parents, outcomes that related to the impact of the whole family, such as partner, siblings, and older children had special significance to parents. The systematic review found that no quantitative studies measuring the impact on children or parenting, yet many parents spoke of the significant psychological impact that the stillbirth had on their children [3]. Four families introduced their children to their baby and involved them in the memory making process. There is mixed evidence on the psychological impact of seeing, holding and spending time with the baby on parents, with conflicting evidence on whether there is a positive or negative effect on PTSD or depression [3]. Further research is required to ascertain the psychological consequences of involving older children in the memory making process, and consideration will have to be made on the feasibility of measuring longer-term outcomes involving family members.

Outcomes connected to the future family were also important to parents. The interviews identified some novel outcomes relevant to planning a subsequent pregnancy including perceived support for planning a subsequent pregnancy, reassurances about fertility and requiring fertility treatment and support. The previous systematic review identified no research studies on interventions or outcomes related specifically to planning a subsequent pregnancy or pre-pregnancy counselling [3]. Findings from these interviews appear to suggest that additional pre-pregnancy support would be appreciated by parents considering planning a subsequent pregnancy following their stillbirth.

Seventy-nine new outcomes were identified specifically related to a subsequent pregnancy, that have not been measured by researchers in the systematic review. The overall concern of the parents who went on to have subsequent pregnancies was the survival of the baby, and therefore a core outcome should acknowledge this appropriately. A range of outcomes were identified from the interviews including medical, social, psychological and experiential outcomes, whilst previous research on care in a subsequent pregnancy has focused on medical outcomes as opposed to the full range of possible outcomes. When comparing with the systematic review, additional outcomes were also identified from the interviews relevant to the maternal and fetal health, hospital resource use (e.g., additional hospital appointments, additional scan appointments) outcomes, bereavement support and care that recognises previous stillbirth [3].

Several outcomes relevant to multiple pregnancy were identified from the interviews. In 2021, a Confidential Enquiry into perinatal deaths in twin pregnancies was published. The report found that 64% of mothers received poor care following the death of their baby or babies and only one third of parents were offered counselling [55]. Outcomes identified from this research could help to evaluate improved bereavement care for families who experience a stillbirth in a multiple pregnancy.

## Conclusion

Finally, this in-depth qualitative study of the lived experience of stillbirth has identified a wide range of novel outcomes, providing a comprehensive and nuanced understanding of the outcomes that matter most to families. By translating the lived experience of stillbirth into potential outcomes, a broad range of impacts were identified, many of which have been overlooked and under-measured in previous stillbirth care research. Importantly, the outcomes span clinical, psychological, social and future pregnancy domains demonstrating the need for a holistic approach to evaluating care impacts. These outcomes have enriched the long-list of outcomes in the next stage of developing a core outcome set for stillbirth care research, ensuring that the priorities and voices of bereaved parents are embedded in final core outcome set [56]. A core outcome set grounded in parental and family insights will help improve the relevance and impact of future research for more effective and compassionate care after stillbirth.

## Supporting information

**S1 Table. Information power dimensions for stillbirth care research interviews.**
(DOCX)

**S2 File. Topic Guide.**
(DOCX)

**S3 Fig. Flow chart of data collection, coding and analysis process for qualitative interviews.**
(DOCX)

**S4 Table. Strategies adopted in study to increase trustworthiness and rigor.**
(DOCX)

**S5 Table. Outcomes identified from interviews.**
(DOCX)

**S6 File. Supplementary quotes to illustrate outcomes and experiences.**
(DOCX)

## Acknowledgments

We would like to thank the families who took the time to participate in interviews and openly shared their experiences and views with us. In addition, we would like to thank the members of the iCHOOSE parent and public involvement group for their assistance in the research.

iCHOOSE Collaborative Group Members

Lead author: Dr Danya Bakhbakhi. Email: danya.bakhbakhi@bristol.ac.uk

Abi Merriel, University of Liverpool; Vicki Flenady, Mater Research Institute – The University of Queensland (MRI-UQ); Soo Downe, University of Central Lancashire; Pauline Slade, University of Liverpool; Lisa Thorne, NHS Devon; Aleena Wojcieszek, Mater Research Institute – The University of Queensland (MRI-UQ); Heloisa de Oliveira Salgado, University of Sao Paulo; Lindsey Wimmer, Star Legacy Foundation; Danielle Pollock, JBI, University of Adelaide; Neelam Aggarwal,

Post Graduate Institute of Medical Education and Research, Chandigarh, India; Susannah Hopkins Leisher, University of Utah School of Medicine and International Stillbirth Alliance; Mehali Patel, Sands Charity, United Kingdom.

## Author contributions

**Conceptualization:** Christy Burden, Dimitrios Siassakos, James MN Duffy, Heatherjane Dangerfield, Lisa Hinton, iCHOOSE Collaborative Group.

**Data curation:** Danya Bakhbakhi, Christy Burden, Anna Davies, Mary Lynch, Laura Timlin, Maggie Redshaw, Lisa Hinton, iCHOOSE Collaborative Group.

**Formal analysis:** Danya Bakhbakhi, Christy Burden, Anna Davies, Abigail Fraser, James MN Duffy, Maggie Redshaw, Alexander EP Heazell, Lisa Hinton, iCHOOSE Collaborative Group.

**Funding acquisition:** Danya Bakhbakhi, Laura Timlin, Lisa Hinton, iCHOOSE Collaborative Group.

**Investigation:** Danya Bakhbakhi, Christy Burden, Anna Davies, Abigail Fraser, Dimitrios Siassakos, Mary Lynch, Laura Timlin, Heatherjane Dangerfield, Lisa Hinton, iCHOOSE Collaborative Group.

**Methodology:** Danya Bakhbakhi, Christy Burden, Anna Davies, Abigail Fraser, Dimitrios Siassakos, Laura Timlin, James MN Duffy, Maggie Redshaw, Heatherjane Dangerfield, Alexander EP Heazell, Lisa Hinton, iCHOOSE Collaborative Group.

**Project administration:** Danya Bakhbakhi, Christy Burden, iCHOOSE Collaborative Group.

**Resources:** Danya Bakhbakhi, iCHOOSE Collaborative Group.

**Software:** Danya Bakhbakhi, Anna Davies, iCHOOSE Collaborative Group.

**Supervision:** Christy Burden, Anna Davies, Abigail Fraser, Dimitrios Siassakos, Maggie Redshaw, Alexander EP Heazell, Lisa Hinton, iCHOOSE Collaborative Group.

**Validation:** Danya Bakhbakhi, Lisa Hinton, iCHOOSE Collaborative Group.

**Visualization:** Danya Bakhbakhi, Christy Burden, Dimitrios Siassakos, Lisa Hinton, iCHOOSE Collaborative Group.

**Writing – original draft:** Danya Bakhbakhi, Lisa Hinton, iCHOOSE Collaborative Group.

**Writing – review & editing:** Danya Bakhbakhi, Christy Burden, Anna Davies, Abigail Fraser, Dimitrios Siassakos, Mary Lynch, James MN Duffy, Maggie Redshaw, Heatherjane Dangerfield, Alexander EP Heazell, Lisa Hinton, iCHOOSE Collaborative Group.

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
