## [Decision Letter · Decision Letter 0]

17 Jun 2025

PONE-D-25-09270What outcomes are important to families with a lived experience of stillbirth? A qualitative study to inform the development of a core outcome set for stillbirth carePLOS ONE

Dear Dr. Bakhbakhi,

Thank you for submitting your manuscript to PLOS ONE. After careful consideration, we feel that it has merit but does not fully meet PLOS ONE’s publication criteria as it currently stands. Therefore, we invite you to submit a revised version of the manuscript that addresses the points raised during the review process.

A rebuttal letter that responds to each point raised by the academic editor and reviewer(s). You should upload this letter as a separate file labeled ‘Response to Reviewers’.A marked-up copy of your manuscript that highlights changes made to the original version. You should upload this as a separate file labeled ‘Revised Manuscript with Track Changes’.An unmarked version of your revised paper without tracked changes. You should upload this as a separate file labeled ‘Manuscript’.

We look forward to receiving your revised manuscript.

Kind regards,

Marcia Leonardi Baldisserotto, Ph.D

Academic Editor

PLOS ONE

Journal Requirements:

1. Please ensure that your manuscript meets PLOS ONE’s style requirements, including those for file naming. The PLOS ONE style templates can be found at

[This report is an independent research arising from a doctoral fellowship (DRF) supported by the National Institute for Health Research (NIHR DRF-2017-10-130). The views expressed in this publication are those of the authors and not necessarily those of the National Health Service, the National Institute for Health Research or the Department of Health.].

3. Thank you for stating the following in your manuscript:

[This report is independent research arising from a doctoral fellowship (DRF) supported by the National Institute for Health Research (NIHR DRF-2017-10-130). The views expressed in this publication are those of the authors and not necessarily those of the National Health Service, the National Institute for Health Research or the Department of Health.]

[This report is an independent research arising from a doctoral fellowship (DRF) supported by the National Institute for Health Research (NIHR DRF-2017-10-130). The views expressed in this publication are those of the authors and not necessarily those of the National Health Service, the National Institute for Health Research or the Department of Health.]

[DB is Trainee Scientific Editor for BJOG.  AEPH is the Director of the Tommy’s Stillbirth Research Centre, University of Manchester which receives grant funding for stillbirth research.].

5. In the online submission form, you indicated that [All relevant data are within the manuscript and its Supporting Information files. Full data cannot be shared publicly due to confidentiality and GDPR. Data are available from the University of Oxford for researchers who meet the criteria for access to confidential data.].

6. Please amend the manuscript submission data (via Edit Submission) to include author Lisa Hinton.

7. One of the noted authors is a group or consortium [iCHOOSE Collaborative Group]. In addition to naming the author group, please list the individual authors and affiliations within this group in the acknowledgments section of your manuscript. Please also indicate clearly a lead author for this group along with a contact email address.

8.  Your ethics statement should only appear in the Methods section of your manuscript. If your ethics statement is written in any section besides the Methods, please move it to the Methods section and delete it from any other section. Please ensure that your ethics statement is included in your manuscript, as the ethics statement entered into the online submission form will not be published alongside your manuscript.

9. Please include a separate caption for each figure in your manuscript.

10. Please include captions for your Supporting Information files at the end of your manuscript, and update any in-text citations to match accordingly. Please see our Supporting Information guidelines for more information: http://journals.plos.org/plosone/s/supporting-information.

11. Please include your tables as part of your main manuscript and remove the individual files. Please note that supplementary tables should remain as separate "supporting information" files.

Reviewers' comments:

Reviewer's Responses to Questions

**Comments to the Author**

1. Is the manuscript technically sound, and do the data support the conclusions?

Reviewer #1: Partly

Reviewer #2: Partly

2. Has the statistical analysis been performed appropriately and rigorously? 

Reviewer #1: N/A

Reviewer #2: N/A

3. Have the authors made all data underlying the findings in their manuscript fully available?

Reviewer #1: Yes

Reviewer #2: Yes

4. Is the manuscript presented in an intelligible fashion and written in standard English?

Reviewer #1: Yes

Reviewer #2: Yes

5. Review Comments to the Author

Reviewer #1: Revision of the manuscript PONE-D-25-09270

What outcomes are important to families with a lived experience of stillbirth? A qualitative study to inform the development of a core outcome set for stillbirth care

I would like to begin by congratulating the authors on addressing such an important and sensitive topic. The focus on identifying outcomes that are important to families with lived experience of stillbirth is highly relevant both to the advancement of research in this area and to improving care practices for families affected by such a devastating event. The study’s aim—to inform the development of a core outcome set for stillbirth care based on the perspectives of families—is both timely and essential, ensuring that research and interventions in this field are aligned with the needs and priorities of those most directly impacted.

That said, despite the relevance and potential contribution of the manuscript, there are several major issues that require careful revision before the manuscript can be considered for publication. These issues pertain particularly to the framing of the study within existing literature, the clarity of methodological procedures, and the depth of analysis and discussion of the results.

Contextualization and Literature Review

The authors argue that their qualitative study identified potential care outcomes, the majority of which had not been previously reported in stillbirth care studies. This is a crucial contribution. However, the manuscript would benefit from a more thorough integration of prior literature on outcomes related to stillbirth care. While the authors refer readers to complementary sources for further context, given the centrality of this point to the value of the study, the introduction should provide a more self-contained and explanatory overview of what is already known in the field. Doing so would better contextualize the originality and significance of the study’s findings, and more clearly highlight the added value of the present work.

Methodological Clarity

Further clarification is needed regarding the sampling and recruitment processes. For instance:

• How were different family members contacted?

• Who facilitated these contacts?

• Were all interviews conducted individually?

• Despite outreach to various family members, only 26 mothers and 12 fathers participated. Were other relatives invited? If so, how many, and what reasons were given for declining to participate?

• Beyond the eligibility criteria described, were there any additional exclusion criteria?

The description of data collection and analysis also requires additional detail. It is unclear how theoretical saturation was determined—was it based on the outcomes themselves or on the codes within these outcomes? Similarly, when the authors state that 298 out of 343 potential outcomes had not been previously reported, is this count based on final outcomes or on individual codes?

Results Presentation

In the results section, while the outcome domains are described, it would greatly strengthen the manuscript to indicate how many participants mentioned each outcome. This would help readers gauge the relative importance or salience of each domain.

Specific clarifications are also needed in several areas:

• Under Care Experience, the authors write: “A number of parents were concerned about the emotional impact that providing stillbirth care had on staff and thus this was identified as an outcome.” This statement is unclear. What exactly did participants mean by this, and how does it translate into an outcome of care?

• Regarding outcomes in the context of multiple pregnancies, how many participants experienced a stillbirth in this specific context?

• In the Grief Outcomes section, the authors note differences in how men and women grieve. What were these differences, and how were they manifested?

• Within Mental Health Outcomes, the text refers to various types of interventions. Were these interventions actually received by participants, or were they interventions the participants wished they had received?

• The same question applies to the Social Outcomes section—are these based on experienced support or unmet needs?

• Concerning Emotional Health Outcomes, where participants mentioned positive emotions such as optimism and pride, it would be important to clarify what these emotions were associated with (e.g., personal growth, support received, subsequent parenting experiences).

• The Partner and Family Relationship Outcomes domain would benefit from further detail regarding how relationships were affected. The current description is too vague.

• For Outcomes Related to Older Children, was the impact described as positive, negative, or mixed? More concrete detail is needed.

• The Outcomes Relevant to Subsequent Children domain also requires further elaboration in order to understand the full scope and significance of the findings.

Discussion and Interpretation

The discussion section appears underdeveloped and overly focused on narrow or specific points. A more comprehensive discussion of the broad range of outcome domains that emerged from the interviews is warranted. In particular, I would have appreciated a deeper comparative analysis between the findings of this qualitative study and those of the authors’ previously conducted systematic review. The discrepancy between these two sources should be examined further: how do the authors explain this divergence? Such an analysis could shed light on potential gaps between research, practice, and the actual needs and perspectives of affected individuals.

In the Strengths and Limitations section, the authors note that a particular effort was made to ensure diversity of experiences, with participants having experienced stillbirths as recently as one year ago and as far back as 50 years. While this breadth is indeed a strength, it would be valuable to explore whether the timing of the experience influenced the outcomes reported. Similarly, did mothers and fathers report different outcomes? Were any other sociodemographic or contextual differences observed in the nature or frequency of outcomes?

Addressing these points would allow the authors to offer more tailored insights for designing research and interventions that are responsive to the varied needs of specific subgroups within this population.

Conclusion

In sum, this manuscript represents an important and timely contribution to stillbirth research, with the potential to meaningfully inform both policy and clinical practice. However, several major issues must be addressed—particularly in terms of contextual framing, methodological transparency, and the breadth and depth of discussion—before this manuscript can be considered ready for publication.

Thank you for the opportunity to review this important work.

Reviewer #2: I congratulate the authors and their interest in these topics, which are so necessary and pertinent. Parents and families who experience stillbirth need special attention and professional care. The article is well organized and adequate in the way the results are presented. The themes and categories are well represented and are in line with recently published studies.

However, some improvements could be made to make the study clearer and more objective. Namely, in introduction, method and resuls.

6. PLOS authors have the option to publish the peer review history of their article (what does this mean?). If published, this will include your full peer review and any attached files.

Reviewer #1: No

Reviewer #2: No

---

## [Author Response · Author response to Decision Letter 1]

26 Aug 2025

I would like to begin by congratulating the authors on addressing such an important and sensitive topic. The focus on identifying outcomes that are important to families with lived experience of stillbirth is highly relevant both to the advancement of research in this area and to improving care practices for families affected by such a devastating event. The study’s aim—to inform the development of a core outcome set for stillbirth care based on the perspectives of families—is both timely and essential, ensuring that research and interventions in this field are aligned with the needs and priorities of those most directly impacted.

Thank you for this compliment.

That said, despite the relevance and potential contribution of the manuscript, there are several major issues that require careful revision before the manuscript can be considered for publication. These issues pertain particularly to the framing of the study within existing literature, the clarity of methodological procedures, and the depth of analysis and discussion of the results.

We have addressed your feedback by revising the manuscript as stated below. We hope this improves the framing of the study within the existing literature, clarity and depth of the results.

The authors argue that their qualitative study identified potential care outcomes, the majority of which had not been previously reported in stillbirth care studies. This is a crucial contribution. However, the manuscript would benefit from a more thorough integration of prior literature on outcomes related to stillbirth care. While the authors refer readers to complementary sources for further context, given the centrality of this point to the value of the study, the introduction should provide a more self-contained and explanatory overview of what is already known in the field. Doing so would better contextualize the originality and significance of the study’s findings, and more clearly highlight the added value of the present work.

Thank you for this feedback, we have now inserted more detail about the types of outcomes reported in previous studies to give more context. Please see page 5 lines 9 to 15.

“Outcomes reported in previous studies of stillbirth care, have primarily focused on medical outcomes, negating the wider impacts of stillbirth on parents4. For example, findings from the same systematic review found no studies on labour and birth care reporting psychological outcomes such as grief or anxiety. Similarly, no studies evaluating bereavement hospital care considered outcomes relating to parents’ role, for example, returning to work or caring for older children”

Further clarification is needed regarding the sampling and recruitment processes. For instance:

1. How were different family members contacted?

2. Who facilitated these contacts?

3. Were all interviews conducted individually?

4. Despite outreach to various family members, only 26 mothers and 12 fathers participated. Were other relatives invited? If so, how many, and what reasons were given for declining to participate?

5. Beyond the eligibility criteria described, were there any additional exclusion criteria?

More information about the sampling and the recruitment is included

1. Interested family members emailed the research team to arrange a potential interview (Page 8 line 9-10).

2. DB (project lead) facilitated recruitment into the study (Page 8 line 10).

3. Parents were interviewed individually or jointly (with partner or family member). (Page 9 line 2).

4. Yes, other relatives were invited including bereaved siblings and grandparents, due to the nature of recruitment approach we are unable to ascertain the total number of family members invited to participate and therefore a response rate or reasons for declining is not documented. (Page 8 line 7-11). Two bereaved siblings participated in the interviews. (Page 11 line 3).

5. No further exclusion criteria were applied.

The description of data collection and analysis also requires additional detail. It is unclear how theoretical saturation was determined—was it based on the outcomes themselves or on the codes within these outcomes? Similarly, when the authors state that 298 out of 343 potential outcomes had not been previously reported, is this count based on final outcomes or on individual codes?

Thank you for this comment. Theoretical saturation was determined when no new outcomes or themes emerged (not codes). More detail is provided in the manuscript Page 8 & 9 lines 20-21 & 1&2

“Recruitment and interviews continued until it was considered that theoretical saturation was approaching (i.e. no new themes or outcomes emerged). To ascertain if the sample size was sufficient, dimensions of information power were also considered including; study aim, sample specificity, use of established theory, quality of dialogue and analysis strategy.”

The final list was based on final outcomes not individual codes. See supporting information 5 and figure 1. Also, page 10 lines 15-18.

“Codes were grouped (where relevant) and experiences translated into outcomes through discussion with the research team. Similar outcomes were arranged into outcome domains, which were then organised into four overarching care outcome themes. See Figure 1 and S3: Figure 1. Outcomes generated were cross-referenced against the outcomes extracted from our systematic review on interventions and outcomes in stillbirth care research. “

In the results section, while the outcome domains are described, it would greatly strengthen the manuscript to indicate how many participants mentioned each outcome. This would help readers gauge the relative importance or salience of each domain.

Thank you. Figure 2 now contains number of interviews mentioning each outcome domain.

Under Care Experience, the authors write: “A number of parents were concerned about the emotional impact that providing stillbirth care had on staff and thus this was identified as an outcome.” This statement is unclear. What exactly did participants mean by this, and how does it translate into an outcome of care?

Thank you, this statement has now been revised and expanded for more context. Page 13 line 4-6.

“A number of parents were concerned about the impact of providing stillbirth care on healthcare professionals; one parent described a midwife handing in her notice another recalled a student midwife “crying floods of tears” after her baby was born.”

Regarding outcomes in the context of multiple pregnancies, how many participants experienced a stillbirth in this specific context? Three sets of parents who had experienced a stillbirth in a multiple pregnancy and a surviving twin sibling were interviewed. This is now included on Page 13 line 16-17.

In the Grief Outcomes section, the authors note differences in how men and women grieve. What were these differences, and how were they manifested? Thank you, have now expanded the section on grief outcomes to include more about the differences in how men and women grieve. Page 14 Line 4-13.

“Common feelings related to overwhelming grief were shock, despair, emptiness, pain, feeling lost, hopelessness, anger and denial. As part of their grief many parents, described feeling guilty and a sense of failure. Mothers described blaming themselves for the death of their baby or wondering if they could have prevented the and so the investigations to understand why a baby died were instrumental in helping to resolve these feelings. Fathers described keeping their feelings to themselves or dealing with their grief at a later point compared with their partners, as they focused on practical tasks such as arranging the funeral and supporting their family. One father did not “grieve properly” in the first year which led to panic attacks and physical symptoms. Fathers often were not offered support by healthcare professions or did not seek support until many years later to help process their grief.”

Within Mental Health Outcomes, the text refers to various types of interventions. Were these interventions actually received by participants, or were they interventions the participants wished they had received? These were interventions received by participants. The following sentences have been revised to provide more clarity. Page 14 Lines 17-20.

“Types of mental health intervention received included cognitive behavioural therapy, mindfulness, pharmacological treatment and eye movement desensitisation and reprocessing (EMDR) for PTSD. Experiences of counselling was discussed by most and reported to have many mental health benefits...”

The same question applies to the Social Outcomes section—are these based on experienced support or unmet needs? These were types of coping support used by parents to help their feelings of isolation. The following sentence has been revised on Page 16 line 1.

“Several coping interventions were utilised by parents, to help alleviate the feeling of isolation...”

Concerning Emotional Health Outcomes, where participants mentioned positive emotions such as optimism and pride, it would be important to clarify what these emotions were associated with (e.g., personal growth, support received, subsequent parenting experiences). We have clarified what positive emotions were associated with on Page 15 line 8-10.

“Conversely positive emotions were also described, such the sense of pride they felt giving birth to their baby and seeing them for the first time. One parent recalled feeling more optimistic after psychological therapy and another had more hope in a subsequent pregnancy when they had specialist care from their hospital.”

The Partner and Family Relationship Outcomes domain would benefit from further detail regarding how relationships were affected. The current description is too vague.

This domain is further expanded on Page 16 Lines 11-25 and Page 17 Line 1-2 to detail further on how relationships were affected.

“Parents reported varying degrees in which their relationships with their partner were affected following the stillbirth. Several described the death of their baby strengthening their relationship and bringing them closer together as a couple. One mother felt that she and her husband dealt with things differently to begin with, however talking about their baby girl helped their relationship, this was facilitated by relationship counselling for some parents. For others it had a negative impact leading to relationship difficulties, arguments, separation or divorce. Trying to become pregnant again soon after the stillbirth led to further challenges and “pressure” to some relationships, particularly when it took longer than expected to conceive.

The impact on relationships in the family were important. According to those interviewed, family members such as grandparents and siblings grieved for the baby and found the emotional impact difficult. Having support and a family network helped many parents with their grief. Friends were also important as a source of support in challenging times and many appreciated people wanting to spend time with them and talk about their stillborn baby. Although friends and family wanted to offer support, sometimes they did not know what to say or do. This left some parents feeling very alone. Some parents described losing friendships when friends were not there for them. Others gained friendships after the stillbirth or found comfort in friends who had also experienced the loss of a baby in pregnancy.”

For Outcomes Related to Older Children, was the impact described as positive, negative, or mixed? More concrete detail is needed. More detail is included on the impact on older children on Page 17 Line 5-22.

“All parents felt it was important that their children were aware of what happened. However, they differed on their preferences on whether to introduce their baby to their children. Some parents decided to not allow their children to meet the baby, either due to the overwhelming shock or the perception that the children were too young at the time to cope. Whilst other older children, decided to meet the baby out of their own choice. Including the children in the memory making process, allowing them to hold, spend time and take photographs was a positive experience for one family, however their six-year-old was upset when they had to say goodbye to their sibling.

Some parents reported no significant effects of the bereavement on their children, whilst others described more long-lasting negative impacts on their mental health and behaviour, including separation anxiety and stress. In the short term, some parents described finding it difficult to concentrate on their children. Whilst for others, older children provided a distraction, allowing some parents to keep going despite their grief. Parents wanted to try and keep everything as normal as possible for their children, by keeping their daily routines. Parents described being more anxious, cautious or over-protective following the stillbirth. Parents described seeking support for children in a variety of ways, including from healthcare professionals such as the bereavement midwife, child bereavement charities, counselling, school or support books written to help parents discuss the stillbirth with their children.”

The Outcomes Relevant to Subsequent Children domain also requires further elaboration in order to understand the full scope and significance of the findings. Further elaboration is included in the outcomes relevant to subsequent children domain. Page 19 Line 8 to 13.

“Some mothers described a delayed grief reaction or postpartum depression which made it hard for them to bond with their new baby in a subsequent pregnancy. A few mothers recalled feeling very frightened and worried about whether their baby would survive the newborn period because of experiencing a stillbirth in the past. This led to more attendances to the doctors due to anxiety over their baby’s health. Having empathic support with parenting from care professionals such as the health visitor was seen as crucial to help them cope.”

The discussion section appears underdeveloped and overly focused on narrow or specific points. A more comprehensive discussion of the broad range of outcome domains that emerged from the interviews is warranted. In particular, I would have appreciated a deeper comparative analysis between the findings of this qualitative study and those of the authors’ previously conducted systematic review. The discrepancy between these two sources should be examined further: how do the authors explain this divergence? Such an analysis could shed light on potential gaps between research, practice, and the actual needs and perspectives of affected individuals. Thank you further in-depth discussion is including on the broad range of outcomes including, grief (Page 22 Lines 3-10, outcomes related to the whole family (Page 23 Lins 7-16), future family (Page 23 Lines 18-24), subsequent pregnancy (Page 23 Lines 26-28 & Page 24 1-7). Divergences and gaps in research are discussed.

“All parents interviewed experienced grief in one form or another, yet their lived experience of this grief was individual. Grief alone is not a unique outcome several studies have measured grief when evaluating care after stillbirth4,34–39. What was most interesting was the multitude of facets of grief that could be measured as identified from the analysis of the interviews, for example overwhelming grief, coping with grief, feelings self-blame, guilt and failure. Any number of these could be measured in a future study, and as such we need to understand what is most important to measure from the perspectives of parents, healthcare professionals and researchers, so as we are able to use the correct measurement tools.

For parents, outcomes that related to the whole family were imperative. Measuring the outcomes related to the impact of the whole family, such as partner, siblings, older children had special significance to parents. The systematic review found that no quantitative studies measuring the impact on children or parenting, yet many parents spoke of the significant psychological impact that the stillbirth ha

---

## [Decision Letter · Decision Letter 1]

29 Sep 2025

PONE-D-25-09270R1What outcomes are important to families with a lived experience of stillbirth? A qualitative study to inform the development of a core outcome set for stillbirth carePLOS ONE

Dear Dr. Bakhbakhi,

Thank you for submitting your manuscript to PLOS ONE. After careful consideration, we feel that it has merit but does not fully meet PLOS ONE’s publication criteria as it currently stands. Therefore, we invite you to submit a revised version of the manuscript that addresses the points raised during the review process.

We look forward to receiving your revised manuscript.

Kind regards,

Marcia Leonardi Baldisserotto, Ph.D

Academic Editor

PLOS ONE

Journal Requirements:

Reviewers' comments:

Reviewer's Responses to Questions

**Comments to the Author**

1. If the authors have adequately addressed your comments raised in a previous round of review and you feel that this manuscript is now acceptable for publication, you may indicate that here to bypass the “Comments to the Author” section, enter your conflict of interest statement in the “Confidential to Editor” section, and submit your "Accept" recommendation.

Reviewer #1: All comments have been addressed

Reviewer #2: All comments have been addressed

2. Is the manuscript technically sound, and do the data support the conclusions?

Reviewer #1: Yes

Reviewer #2: Yes

3. Has the statistical analysis been performed appropriately and rigorously? 

Reviewer #1: N/A

Reviewer #2: N/A

4. Have the authors made all data underlying the findings in their manuscript fully available?

Reviewer #1: No

Reviewer #2: Yes

5. Is the manuscript presented in an intelligible fashion and written in standard English?

Reviewer #1: Yes

Reviewer #2: Yes

6. Review Comments to the Author

Reviewer #1: Review Letter for Manuscript PONE-D-25-09270_R1

Title: What outcomes are important to families with lived experience of stillbirth? A qualitative study to inform the development of a core outcome set for stillbirth care

Thank you for the opportunity to review the revised version of this important manuscript. I would like to sincerely congratulate the authors for the evident effort they put into revising the article and for their careful attempt to incorporate the suggestions provided in the first round of review. These efforts have clearly resulted in a significant improvement of the manuscript and in a clearer and more accessible presentation of the study for readers.

Nevertheless, I would like to draw the authors’ attention to a few minor issues that could further strengthen the manuscript:

On page 9, line 6, there appears to be a typo: I believe a comma (,) rather than a semicolon (;) should follow “gestation or more;”.

On page 10, the authors note that interviews were conducted either individually or together with the partner or another family member. I would kindly suggest including some brief considerations regarding whether this may have influenced participants’ behavior during the interviews, how coding was managed when information came from different participants within the same interview, and whether such circumstances had any impact on the results.

In the Results section, when presenting the outcome themes and domains, there are still some instances of vague expressions such as “several parents,” “several mothers,” “a number of parents,” “many parents,” or “by most.” I encourage the authors to replace these with objective quantitative indicators, specifying how many participants actually made those references.

In the Discussion section, on page 26, I recommend that the authors not only emphasize the importance of including both parents but also discuss the relevance of allowing participation of other family members, highlighting the value this broader perspective may bring.

I suggest introducing the final paragraph before the Conclusion with the phrase “Finally,” to strengthen the flow of the argument.

Lastly, the article would benefit from a more comprehensive Conclusion section, providing greater detail and reinforcing the contributions and value of this work.

In summary, this is a much-improved manuscript that makes a valuable contribution to the field. For the reasons outlined above, I am favorable to its publication in its current form, once the minor issues highlighted are addressed.

Reviewer #2: (No Response)

7. PLOS authors have the option to publish the peer review history of their article (what does this mean?). If published, this will include your full peer review and any attached files.

Reviewer #1: No

Reviewer #2: No

---

## [Author Response · Author response to Decision Letter 2]

1 Apr 2026

Centre for Academic Women’s Health

School of Translational Health Sciences

University of Bristol

Southmead Hospital

Westbury on Trym

Bristol

BS10 5NB

United Kingdom

10th March 2026

Dear PLOS One,

Please find enclosed our second revised manuscript entitled “What outcomes are important to measure for stillbirth care research? A qualitative study of the lived experience of stillbirth to inform the development of a core outcome set” by Bakhbakhi et al. We hope our amendments make this manuscript suitable for publication in PLOS One. Please see response to reviewers for all edits to manuscript and responses to reviewers.

Please see response to reviewer’s comments below:

1. On page 9, line 6, there appears to be a typo: I believe a comma (,) rather than a semicolon (;) should follow “gestation or more;”.

This has been amended accordingly in the manuscript.

2. On page 10, the authors note that interviews were conducted either individually or together with the partner or another family member. I would kindly suggest including some brief considerations regarding whether this may have influenced participants’ behavior during the interviews, how coding was managed when information came from different participants within the same interview, and whether such circumstances had any impact on the results.

Thank you for this helpful suggestion.

We have added this as a potential limitation in the discussion on Page 26 Line 14-15. “Furthermore, some interviews were conducted together with the partner, this may or may not have influenced what a parent chose to share or emphasise, and we note this as potential limitation.”

We have also clarified the coding approach in the methods as follows on Page 11 Line 21-23

“Each participant’s contributions/outcomes, whether from the bereaved mother, partner or family member were coded individually and each quotation presented in the results is attributed to the specific parent who said it. ”

In addition, we have updated Table 2. Outcome themes and domains with illustrative participant quotes to specify which parent or family member contributed to each quote, providing greater transparency.

3. In the Results section, when presenting the outcome themes and domains, there are still some instances of vague expressions such as “several parents,” “several mothers,” “a number of parents,” “many parents,” or “by most.” I encourage the authors to replace these with objective quantitative indicators, specifying how many participants actually made those references.

Thank you for highlighting this. We have carefully revised the Results section to replace vague descriptors (e.g., “several,” “many,” “a number of parents”) with specific counts indicating how many participants raised each outcome. This provides greater clarity and transparency regarding the distribution of views.

At the same time, we have avoided providing exact numerical figures in every instance to remain aligned with established qualitative research principles, which emphasise depth of understanding rather than quantification. As noted by Green and Thorogood in Qualitative Methods for Health Research (Sage, 2018), qualitative enquiry aims to explore “what, how, or why” rather than measure phenomena, and the data generated are typically in the form of words rather than numbers. We therefore report precise participant numbers where appropriate while maintaining fidelity to qualitative methodology.

4. In the Discussion section, on page 26, I recommend that the authors not only emphasize the importance of including both parents but also discuss the relevance of allowing participation of other family members, highlighting the value this broader perspective may bring.

Thank you for this suggestion. We have included this statement in the discussion “An important strength of this study was that an inclusive approach was adopted, including both parents and participation of other family members in the interviews, thus broadening the scope of outcomes identified.”

5. I suggest introducing the final paragraph before the Conclusion with the phrase “Finally,” to strengthen the flow of the argument.

Thank you this has been added to page 30 line 13.

6. Lastly, the article would benefit from a more comprehensive Conclusion section, providing greater detail and reinforcing the contributions and value of this work.

Thank you a more comprehensive conclusion section has been included. See Page 30 line 13-21.

“Finally, this in-depth qualitative study of the lived experience of stillbirth has identified a wide range of novel outcomes, providing a comprehensive and nuanced understanding of the outcomes that matter most to families. By translating the lived experience of stillbirth into potential outcomes, a broad range of impacts were identified, many of which have been overlooked and under-measured in previous stillbirth care research. Importantly, the outcomes span clinical, psychological, social and future pregnancy domains demonstrating the need for a holistic approach to evaluating care. These outcomes have enriched the long-list of outcomes in the next stage of developing a core outcome set for stillbirth care research, ensuring that the priorities and voices of bereaved parents are embedded in final core outcome set. A core outcome set grounded in parental and family insights will help improve the relevance and impact of future research for more effective and compassionate care after stillbirth.”

7. In summary, this is a much-improved manuscript that makes a valuable contribution to the field. For the reasons outlined above, I am favorable to its publication in its current form, once the minor issues highlighted are addressed.

Thank you.

---

## [Editor Report · Decision Letter 2]

5 Apr 2026

What outcomes are important to families with a lived experience of stillbirth? A qualitative study to inform the development of a core outcome set for stillbirth care

PONE-D-25-09270R2

Dear Dr. Bakhbakhi,

We’re pleased to inform you that your manuscript has been judged scientifically suitable for publication and will be formally accepted for publication once it meets all outstanding technical requirements.

Kind regards,

Marcia Leonardi Baldisserotto, Ph.D

Academic Editor

PLOS One
---

## [Editor Report · Acceptance letter]

PONE-D-25-09270R2

PLOS One

Dear Dr. Bakhbakhi,

I'm pleased to inform you that your manuscript has been deemed suitable for publication in PLOS One. Congratulations! Your manuscript is now being handed over to our production team.

Kind regards,

on behalf of

Dr. Marcia Leonardi Baldisserotto

Academic Editor

PLOS One